# Do Service Dogs for Veterans with PTSD Mount a Cortisol Response in Response to Training?

**DOI:** 10.3390/ani11030650

**Published:** 2021-03-01

**Authors:** Emmy A. E. van Houtert, Nienke Endenburg, T. Bas Rodenburg, Eric Vermetten

**Affiliations:** 1Animals in Science and Society, Faculty of Veterinary Medicine, Utrecht University, 3584 CM Utrecht, The Netherlands; n.endenburg@uu.nl (N.E.); t.b.rodenburg@uu.nl (T.B.R.); 2Department of Psychiatry, Leiden University Medical Centre, 2311 EZ Leiden, The Netherlands; e.vermetten@lumc.nl; 3ARQ National Psychotrauma Center, 1112 XE Diemen, The Netherlands; 4Department of MGGZ, Ministry of Defence, 3584 EZ Utrecht, The Netherlands

**Keywords:** AAI, PTSD, service dogs, welfare

## Abstract

**Simple Summary:**

A growing number of people are supported by specialized service dogs. These dogs are highly trained to improve human welfare, yet not much is known about their own welfare. One of the ways in which welfare can be measured is through the expression of stress via the hormone cortisol. In this study, we investigated the level of cortisol in saliva, a measure for physiological stress, in 19 service dogs. We measured cortisol in the dogs’ saliva 15 min after arrival at a training ground, before partaking in a training session for service dogs, after participation in the training session, and after a 45-min free play period. We found no elevated levels of cortisol after the training session. Instead, we found that cortisol had lowered when compared to before the training. Additionally, we found that cortisol was highest 15 min after arriving at the training round and after 45 min of free play. This led to the conclusion that dogs in our study did not seem to have a stress response in response to participation in the training.

**Abstract:**

Only a few studies have investigated the welfare of animals participating in animal-assisted interventions (AAIs). Most of these studies focus on dogs in therapeutic settings. There are, however, also dogs—service dogs—that are employed to continuously support a single human. Because the welfare of these service dogs is important for the sustainability of their role, the aim of this study was to investigate their stress response to service dog training sessions. To do this, we took repeated salivary cortisol samples from dogs who participated in a training session (*n* = 19). Samples were taken just after arrival at the training ground, before training, after training, and after a period of free play. Our results showed that mean cortisol levels in all samples were relatively low (between 1.55 ± 1.10 and 2.73 ± 1.47 nmol/L) compared to similar studies. Analysis further showed that samples taken before and after participation in the training’s session did not differ from one another. Mean cortisol levels in both situations were additionally lower than those upon arrival at the training site and after a period of free play. This led to the conclusion that the dogs in our study did not seem to experience training as stressful.

## 1. Introduction

The relationship between humans and dogs knows a long history. Dogs have assisted humans in a growing array of tasks. These tasks include tracking specific scents [1,2], guarding objects, people, or locations [3], cattle herding, pulling carts, scrap cleaning (and through this, pest and disease control), providing companionship, and providing warmth [4]. As of the 20th century, there has additionally been a growing interest in the development and deployment of specialized dogs to improve individual human health. Perhaps the best known of these dogs is the guide dog for humans with a visual disability. Other examples include dogs for those with a hearing impairment [5], dogs that detect low blood sugar [6], dogs that detect symptoms of epileptic seizure [7], dogs that assist with a physical disability [8], dogs that assist with autism spectrum disorder (ASD) [9,10], and dogs that assist those with a post-traumatic stress disorder (PTSD) [11,12].

Dogs intentionally deployed for the welfare of humans are collectively known as either service dogs or assistance dogs (region-dependent). Their deployment is further considered a form of animal-assisted intervention (AAI), which entails that an animal is used in a (therapeutic) intervention for the improvement of human welfare and/or health. Since the goal of AAI is aiding humans, studies on the topic have mainly focused on the effects that the animals have on the humans they are aiding. Only a few studies and publications have discussed the effect of AAI on animal welfare [13,14,15,16] and even fewer have studied animal welfare in AAI via experimental design.

Most studies that have focused on animal welfare in AAI concentrated on the deployment of dogs in therapeutic settings. They did so primarily through a combination of behavioral assessment through structured observation and the analysis of cortisol samples. The use of heart rate and body temperature is, however, also seen [14,16,17,18,19,20,21,22]. Although it is disputed whether there is a relation between behavioral observations and cortisol measurements in dogs [23,24,25,26,27], both measures have individually been found to be indicative of animal welfare status. Behavioral observation, for example, has been established as a tool to assess arousal or stress in dogs [28,29]. Dogs that are subjected to stressors such as social or spatial restriction are known to perform specific behaviors more often than relaxed dogs. Examples of such behaviors include yawning without other signs of drowsiness, paw lifting, body shaking without a waterlogged fur, and walking around erratically [30,31]. The performance of these behaviors has further been linked to a state of either conflict, confusion, or fear in dogs [32], which can, in turn, be used to determine if an individual dog is either physically or mentally able to cope with the situation it is currently in.

Changes in the concentration of the steroid hormone cortisol have additionally been associated with physiological signs of stress in dogs and other mammals [28,33,34,35], though it deserves mention that heightened cortisol is also a possible sign of positive arousal. Although cortisol can be found in various bodily fluids [36,37,38], one of the less invasive, yet accurate, ways in which to detect it is through a salivary swab [39,40]. Because of this reduced invasiveness, salivary cortisol has become a widely used method to determine both acute [23] and chronic stress [24,30] in dogs. It has additionally given insight into dogs’ recovery process from acute stressors, as demonstrated by Beerda et al. in their study from 1998 [23]. In their study, they found that salivary cortisol in dogs showed a 13- to 20-nmol/L elevation compared to the basal level (mean 6 nmol/L) after the dogs had been exposed to an acute stressor (opening umbrella, sudden shock). The time it took for this peak to appear was between 0 and 30 min following the stressor, which is in line with the time it takes salivary cortisol to reflect plasma cortisol [41].

Beerda et al. [23] additionally found that peak values of salivary cortisol had dropped by half in most dogs 30 min post-stressor and returned to baseline levels after 45–60 min post-stressor. These findings indicate that the observed dogs had a capacity to recover from their encountered stressor and return to baseline values if given time to do so. This capacity to recover from stressors is particularly important for dogs in AAI as they are exposed to potential stressors on a regular basis [13]. Dogs that are used for AAI are, therefore, often pre-selected for their capacity to recover from stressors via a series of temperament tests and behavioral observations. They are additionally specifically trained from a young age to familiarize them with the work they will perform in later life. In theory, therefore, only animals that are both mentally and physically capable of assistance work are employed. To test if this assumption is true, studies such as those by Glenk et al. [14,18] and Clark et al. [16] have evaluated the effect of assistance work on dog welfare in AAI. This was mostly done during therapeutic sessions in which dogs performed assistance work for several individuals one, two, or three times a week. In their studies, Glenk et al. [14,18] and Clark et al. [16] reported no indications of (severe) stress in the dogs after they had assisted in a therapy session, which can be interpreted as meaning that selected and trained dogs are capable of coping with the stressors of assistance work. That is not to say that this conclusion holds true for all dogs in AAI though, as settings and workloads tend to differ between subtypes of AAI. There are, for example, also dogs who assist a single human 24/7 as opposed to several humans during a therapeutic session two to three times a week. This subtype is often referred to as a service dog and has a more unpredictable and more frequent workload than the dogs observed in earlier studies. These dogs too, however, are pre-selected and trained for their work, which should mean they are mentally and physically capable of the work they are asked to perform in a similar manner as dogs used during therapy sessions.

To test if service dogs are capable of handling the tasks they are asked to do during their working life, we wanted to know if they showed physiological signs of stress during their work. Because a service dog’s work is highly variable, however, we instead chose to evaluate dogs during a standardized situation which is similar for each dog. As such, we questioned whether service dogs show physiological signs of stress during a training session for active service dogs (as indicated by heightened salivary cortisol), and if so, whether they can recover from this stress within a time span of 45–60 min. If the dogs do not show a salivary increase after training, it can be argued that they did not experience the training as stressful. If they do show elevation after training yet show a return to baseline values after a recovery period, it can be argued that the dogs are capable of coping with the stressors they experienced during training. Both answers could help to evaluate whether service dogs are properly prepared for the work that is asked from them through their selection and training, or if these procedures need to be re-evaluated for future generations of service dogs.

## 2. Materials and Methods

### 2.1. Subjects

For this study, 19 service dogs were observed. All dogs were trained and licensed service dogs of the Dutch service dog provider “Stichting Hulphond Nederland” and deployed to assist a single military veteran or (ex-)first aid responder with post-traumatic stress disorder (PTSD) (referred to as handler). They had additionally been living with their assigned handler full-time for at least a year and were used to working with them in daily tasks. Among the dogs, 16 were purebred Labrador Retrievers, one a Standard Poodle, one an Airedale Terrier, and one a mix between a Malinois and Labrador Retriever. The male/female ratio was 17/2 (all spayed/neutered), while the age of all dogs was between two and eight years (average 3.9 ± 0.7), as these are the regular working years of a service dog (between training and retirement). To participate in this study, all dogs finally needed to be in good clinical health (as judged by a veterinarian) and were obliged to have had regular (at least four times a year) behavioral monitoring by an animal trainer from the service dog provider during the past year.

### 2.2. Experimental Design

Measurements for this study were taken during one of two collective training days at a service dog training facility of “Stichting Hulphond Nederland”. These training days were part of the service dogs’ ongoing training and primarily serve to help to reinforce trained behaviors on a periodical basis after they have been matched with a handler. They additionally serve as an opportunity to assess the development of the relationship between the dog and the handler. Due to the varying ages of the dogs participating in this study, some dogs were familiar with this form of training while others were not.

Although PTSD service dogs usually work in their own home environment, a collective training session was chosen as a measurement moment to standardize conditions between dogs. During training, dogs had to perform a novel navigation task (such as following a specific path between obstacles). They could only complete this task by communicating with their handler since the dogs did not know the desired route between or around presented obstacles. With this method, the training simulated elements of the service dogs’ work in a controlled environment, namely helping their handler navigate a distracting and often unfamiliar environment while maintaining focus on the handler.

During the collective training, four saliva samples were collected from each dog (*n* = 19). This was done by placing a SalivaBio children swab (Salimetrics, 5001.06 and 5001.05) [42,43] in saliva pooling areas (mouth corners or under the tongue) in the mouth of the dog. In this manner, the swabs could passively absorb saliva for 60 s, while the dog was gently held around the muzzle. To prevent contamination of the samples, the dogs were not given any treats for at least 10 min prior to sampling. After sampling, the dogs were given a treat, however, to reward good behavior. The complete process of sampling was less than four minutes [40] for each sample so as to prevent the procedure from influencing the sample.

Out of the four samples, the first sample was collected 15 min after dogs had arrived at the training ground (T + 15). This was done to measure anticipation stress in the dogs caused by the arrival at the training ground. The dogs were then given 30 min to adapt to their new environment before the second sample was collected at the start of the training exercise (T + 45). During this 30-min gap, the dogs were either interacting with other dogs, walking with their handler, or resting while their handler received instructions for the training.

The third sample was taken, again, 30 min after the dogs had finished their training exercise (T + 75). They were subsequently given 45 min of free time after training, at the end of which the fourth sample was collected (T + 120) (see, also, Figure 1). During these 45 min, dogs were free to either play with other dogs present, play with their handlers, go for a walk with their handlers, or rest.

All samples were subsequently stored at −20 °C until saliva extraction. Extraction was performed by spinning the samples at 3000 rpm for 5 min. This resulted in a clear supernatant of low viscosity. A visual inspection was additionally performed at this stage for any signs of contamination (discoloration). No samples were rejected because of this. Cortisol concentrations were finally measured using a commercially available chemiluminescence immunoassay with high sensitivity (IBL International, Hamburg, Germany). The average intra-assay coefficient was 4%.

### 2.3. Statistical Analysis

Statistical analysis of salivary cortisol levels between all four samples taken during the collective training day was performed in R via Skillings–Mack test for non-parametric paired data with missing data points. Additional analysis of all possible sample pairs was performed via Wilcoxon signed rank test for paired non-parametric data. A Holm–Bonferroni correction was additionally performed on these tests to correct for multiple testing. Mauchly’s and Levene’s tests were finally performed to test for sphericity and equal variance of the dataset, respectively.

### 2.4. Ethical Statement

Ethical advice regarding this study was sought with the university’s resident animal experiment advisory board. Because no invasive measurements were taken, however, the full protocol of this study did not require judgement by the ethical committee.

## 3. Results

### 3.1. Missing Values

Out of the samples collected during the collective training days, the volume of retrieved saliva was sufficient for analysis in 67% of samples. Out of the total 19 dogs, nine had four sufficient samples, six dogs had three sufficient samples, two dogs had two sufficient samples, one dog had one sufficient sample, and one dog had zero sufficient samples. These missing values bring the total amount of successful samples at each time point to *n* = 13 at T + 15, *n* = 16 at T + 45, *n* = 16 at T + 75, and *n* = 14 at T + 120.

### 3.2. Cortisol Levels

The average salivary cortisol level of the dogs at the start of the collective training day (T + 15; *n* = 13) was 2.73 nmol/L (±1.47). At the start of training (T + 45, *n* = 16), this level was 2.28 nmol/L (±1.51). It was 1.65 nmol/mL (±0.64) at the end of training (T + 75, *n* = 16), and finally, 2.33 nmol/L (±0.83, *n* = 14) 45 min after the training session had ended (T + 120; Figure 2).

### 3.3. Statistical Analysis

The Skillings–Mack test statistic was 33.05 (*p* = 0.01, α = 0.05), which indicates that the four measurement points of this study differed from one another. To identify which specific data points caused this result, an additional analysis was performed via Wilcoxon signed rank test between all data points in combination with a Holm–Bonferroni correction. The Wilcoxon signed rank tests indicated the data points T + 15/T + 75 and T + 75/T + 120 to be significantly different (Figure 3). The Holm–Bonferroni correction, however, did not yield significant differences between combinations. This combination of results indicates that T + 15/T + 75 and T + 75/T + 120 might be significantly different, though a Type I error cannot be excluded. All other data point combinations did not differ significantly from one another in both tests. Mauchly’s and Levene’s tests were finally performed to test for sphericity (*p* = 0.49) and equal variance (*p* = 0.10) of the dataset, respectively.

## 4. Discussion

In this study, we questioned whether service dogs would show physiological signs of stress during a service dog training session and, if so, whether they recover from this stress within a time span of 45–60 min. Our results did not show any indication of acute stress experienced due to participation in the training, as salivary cortisol levels before and after training did not differ significantly from each other. This is in line with earlier findings by Glenk et al. [14,18] and Clark et al. [16] in therapy dogs, as they also did not find a significant effect of assistance work on the level of salivary cortisol in assistance dogs.

The cortisol levels retrieved during this study (mean cortisol T + 15 = 2.73, T + 45 = 2.28, T + 75 = 1.65, T + 120 = 2.33 nmol/L) were slightly lower than those found during earlier studies. A meta-analysis by Cobb et al. [44], for example, found an overall mean basal salivary cortisol level of 0.45 ug/dL or 12.42 nmol/L between various dog studies. This, however, included studies with various dog breeds in different situations such as shelter dogs, companion dogs, guide dogs, and laboratory animals. Because of this diversity in breeds and settings, the values calculated by Cobb et al. [44] are potentially not representative of specific subsets (breeds or disciplines) of dogs. A study by Koyama et al. in 2003 [45] in Beagles, for example, found lower values for 24-h salivary cortisol variability of dogs. They found resting cortisol to be fluctuating between 2 and 8 nmol/L, with the interesting remark that no distinct circadian cortisol rhythm seemed to be present in dogs, as it is in most other mammals. A study by Beerda et al. [23] found results in agreement with Koyama et al. [45], as they reported basal cortisol levels to be 6 nmol/L in their dogs (mainly Beagles) in an experimental setting. Because of the above, it could be that the dogs in this study (mainly Labrador Retrievers) had a natural disposition for low cortisol due to their genetic background. A study by Batt et al. [46], however, found salivary cortisol values in guide dogs in training (also mainly Labrador Retrievers) which exceeded the results found in our study and those by Beerda et al. [23] and Koyama et al. [46] (2.07–2.17 ug/dL = 57.11–59.87 nmol/L).

Because of the above, it is more likely that the lower cortisol values in this study were caused by the setting in which it was performed. Within this study, a total of four measurement points were used surrounding a single intervention (the training session). Out of these measurement points, those preceding the intervention were higher than the one following it (significant for T + 15/T + 75). As stated before, this observation is in line with earlier findings by Glenk et al. [14,18] and Clark et al. [16] in therapy dogs, as they also did not find an increasing effect of assistance work on the level of salivary cortisol in assistance dogs after participation in a therapy session. It is, additionally, contrary to results found by van der Borg et al. [47], who found that salivary cortisol did increase relative to pre-intervention levels when dogs were exposed to a stressful situation. They additionally found that cortisol levels lowered again after a 30-min resting period, which is, in turn, in line with the results of Beerda et al. [23].

Because van der Borg et al. [47] noted a possibility for salivary cortisol to increase in a setup comparable to our study, it can be assumed that the lowering of cortisol seen in our study after the start of training is indicative of lowering physiological stress. It can, therefore, be argued that the dogs in our study either did not show signs of physiological stress in response to the training they took part in or that they were able to recover before their salivary cortisol was re-measured (30-min gap). Given that it takes roughly 45–60 min for salivary cortisol in dogs to fully return back to basal levels after encountering a stressor [23,29], the former explanation seems more plausible than the latter.

The possibility of lowered physiological stress in dogs after training is, finally, supported by their salivary cortisol levels shortly after arrival at the training ground and after 45 min of free play. These levels were elevated compared to pre- and post-training levels, which indicates that the dogs experienced more physiological stress or arousal at these time points than during training. This, in turn, also suggests against the possibility of long-term stress-induced Hypothalamic–pituitary–adrenal (HPA) axis downregulation in dogs [48], since this generally reduces cortisol reactivity. In the case of arrival at the training ground, the elevation might have been caused because the dogs encountered unfamiliar surroundings and conspecifics, which might have acted as a stressor or a stimulator. Meeting other dogs could have additionally increased activity in the dogs, which, by itself, is known to increase cortisol levels in animals [49,50]. This last effect would additionally explain the elevation seen after free play, as the dogs were allowed to play with one another on the training field. It might further explain the greater variation in salivary cortisol levels seen at both time points, as not all dogs were equally engaged in play behavior.

All in all, our results, therefore, indicate that the dogs in this study did not experience the training as physiologically stressful. Out of all the time points, the dogs in our study showed the lowest cortisol response with the least variation among them right after they had partaken in training. Combined with the fact that this data point differed significantly from measurements taken right after arrival at the training ground (when dogs might have also experienced stressors), we interpret that the dogs in our study did not experience the training as stressful. As there appeared to be no stress to recover from, it is difficult to draw additional conclusions about the dogs’ capacity to recover from stressors. Given that the dogs did show elevated cortisol immediately after arrival at the training ground, however, which lowered after an acclimatization period (not significant), it could be interpreted that this capacity is present. A note of caution needs to be added that only from nine dogs were samples at all time points available, underlining the need to confirm these results in a larger number of service dogs. We finally conclude that the service dogs in our study did not appear to experience training as physiologically stressful, but instead seemed to be able to cope with the work that was required from them.

## Figures and Tables

**Figure 1 animals-11-00650-f001:**
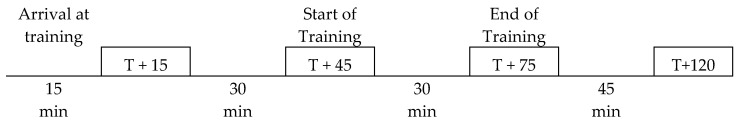
An overview of the different sample moments of this study relative to the arrival of the dogs at the training ground.

**Figure 2 animals-11-00650-f002:**
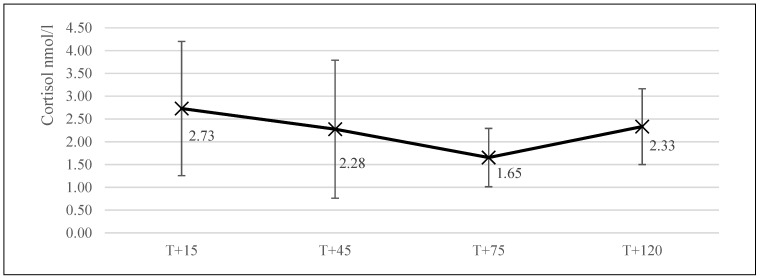
The various mean levels of salivary cortisol (±SD) at the four different sample points for the collective training session (*n* = 13 at T + 15, *n* = 16 at T + 45, *n* = 16 at T + 75, and *n* = 14 at T + 120). The training session took place between T + 45 and T + 75.

**Figure 3 animals-11-00650-f003:**
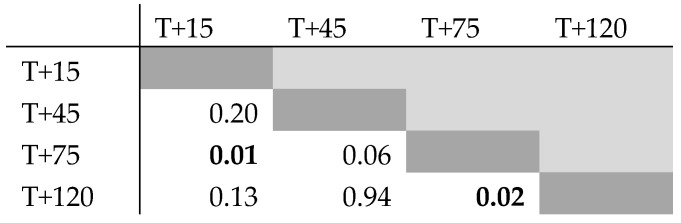
The results of the Wilcoxon signed rank test matrix between all measurement points of this study (*n* = 19). The two measurement moments with a significant (α = 0.05) difference are in **bold**. A Holm–Bonferroni correction showed that these two points are susceptible to Type I error, however.

## Data Availability

The data presented in this study are available on request from the corresponding author.

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
