# Peer review of "Do Service Dogs for Veterans with PTSD Mount a Cortisol Response in Response to Training?"

_animals, 2021, doi:10.3390/ani11030650_

Round 1

Reviewer 1 Report

Line 11: "people are"

Line 24: "so called" is awkward phrasing

line 42: beast of burden?

line 50: typo "with from"

line 79: typo "et al." several times through manuscript

line 98: typo

line 130: "finally obligated" could be re-phrased

line 181: typos

Methods:

Need to provide more specific details of the collective training session (duration) and the free play session, activities location, conspecifics/contraspecifics present. Need to provide details on dogs' familiarity with the training environment and people present. Dogs were already trained (with handler at least a year), was this training on-going, a refresher or just for this study? Were the dogs familiar with this location?

Results:

lines 184-187. Results indicate N=19 however, section 3.1 indicates only 9 dogs had a 4 sufficient samples.  What is the N for each time sample (T+15n T+45, T+75 and T+120).

Figure 2: indicates N=19 but N was not 19 for all samplepoints

Discussion: Data indicates a complete Repeated measure sampling across all 4 samplepoints is 9 dogs. Conclusions need to address this low N, especially with the incredibly high individual variation of cortisol. Assumptions need to be made with caution due to this low N.  Discussion briefly speculates that free play included play with conspecifics, this should be included in the methods.  Authors discuss possible reasons for cortisol levels at each time point but again the N for each timepoint is not given.  The absence of this N makes it impossible to evaluate if the conclusions are fully supported by the data.

Author Response

Line 11: "people are"

Line 24: "so called" is awkward phrasing

line 42: beast of burden?

line 50: typo "with from"

line 79: typo "et al." several times through manuscript

line 98: typo

line 130: "finally obligated" could be re-phrased

line 181: typos

Response:

We would like to thank the reviewer for their sharp eye in detecting the above mentioned typo’s and textual issues. With the exception of the issue in line 11 all issues have been addressed as requested. Regarding the issue in line 11 we think that the use of ‘ is ‘ is grammatically appropriate since the word refers to  the singular ‘ number’.

Methods:

Need to provide more specific details of the collective training session (duration) and the free play session, activities location, conspecifics/contraspecifics present. Need to provide details on dogs' familiarity with the training environment and people present. Dogs were already trained (with handler at least a year), was this training on-going, a refresher or just for this study? Were the dogs familiar with this location?

Response:

We would like to thank the reviewer for addressing this issue. Extra information has been added to the manuscript regarding dog activity during the various phases of this study. Extra information has additionally been added to describe the background of the presented training. (See line 135-150)

Results:

lines 184-187. Results indicate N=19 however, section 3.1 indicates only 9 dogs had a 4 sufficient samples.  What is the N for each time sample (T+15n T+45, T+75 and T+120).

Response:

As per request the specific n for each sample time has been added to the manuscript instead of the overall n=19 (See line 200-202)

Figure 2: indicates N=19 but N was not 19 for all samplepoints

Response:

As per request the specific n for each sample time has been added to the manuscript instead of the overall n=19 (See line 218 and Figure 2)

Discussion: Data indicates a complete Repeated measure sampling across all 4 samplepoints is 9 dogs. Conclusions need to address this low N, especially with the incredibly high individual variation of cortisol. Assumptions need to be made with caution due to this low N. 

Response:

We agree with this point and have added a note of caution just before the final conclusion (line 294-296).

Discussion briefly speculates that free play included play with conspecifics, this should be included in the methods. 

Response:

A mention of the possibility to play with conspecifics during the free play period between T+75 and T+120 has been added to the methodology section as per reviewer request. (line 170-172).

Authors discuss possible reasons for cortisol levels at each time point but again the N for each timepoint is not given.  The absence of this N makes it impossible to evaluate if the conclusions are fully supported by the data.

Response:

As per request the specific n for each sample time has been added to the manuscript instead of the overall n=19 (line 204-207).

Reviewer 2 Report

I really liked the manuscript and consider it an interesting topic that should be investigated extensively.
I would like to make some suggestions:
I wanted to know why these dogs have been specifically selected? Dogs for veterans with PTSD
Do the authors think that the results would be modified with other types of working dogs?
I would like to see more information about dog training sessions
Some information about the ethical authorization of the study should be included

Author Response

I really liked the manuscript and consider it an interesting topic that should be investigated extensively.

Response:

We would like to thank the reviewer for this comment

I would like to make some suggestions:
I wanted to know why these dogs have been specifically selected? Dogs for veterans with PTSD

Response:

This study was performed as part of a larger research project to the influence of PTSD service dogs on veterans with PTSD. In this project both the welfare of animals and humans involved in this form of AAI is evaluated. This study was part of the animals welfare part of the overall project.

Do the authors think that the results would be modified with other types of working dogs?

Response:

Without further study it is difficult to state if the observation of different types of working dogs would produce results different to this study. This would depend on various factors like previous training of the dogs, overall care, temperament selection, breed, and other factors. Similar results are however possible in different settings given the results found by Glenk et al. and Clark et al. (as described in the introduction section of this manuscript).

I would like to see more information about dog training sessions

Response:

As per request extra information about the purpose and content of the training session has been added.

Some information about the ethical authorization of the study should be included

Response:

Ethical advice regarding this study was sought with the university’s resident animal experiment advisory board. Because no invasive measurements were taken however, the full protocol of this study did not require judgement by the ethical committee. This statement has been added to the manuscript. (Line 191-194)